# RAC1B: A Guardian of the Epithelial Phenotype and Protector Against Epithelial-Mesenchymal Transition

**DOI:** 10.3390/cells8121569

**Published:** 2019-12-04

**Authors:** Rabea Zinn, Hannah Otterbein, Hendrik Lehnert, Hendrik Ungefroren

**Affiliations:** 1First Department of Medicine, UKSH, Campus Lübeck, 23552 Lübeck, Germany; rabeazinn@googlemail.com (R.Z.); hannahotterbein@web.de (H.O.); hendrik.lehnert@uni-luebeck.de (H.L.); 2Department of General Surgery, Visceral, Thoracic, Transplantation and Pediatric Surgery, University Hospital Schleswig-Holstein, Campus Kiel, 24105 Kiel, Germany

**Keywords:** RAC1B, RAC1, pancreatic ductal adenocarcinoma, Rho GTPase, signaling, epithelial-mesenchymal transition, extracellular signal-regulated kinase, transforming growth factor-β

## Abstract

The small GTPase Ras-related C3 botulinum toxin substrate 1B (RAC1B) has been shown to potently inhibit transforming growth factor (TGF)-β1-induced cell migration and epithelial-mesenchymal transition (EMT) in pancreatic and breast epithelial cells, but the underlying mechanism has remained obscure. Using a panel of pancreatic ductal adenocarcinoma (PDAC)-derived cell lines of different differentiation stages, we show that RAC1B is more abundantly expressed in well differentiated as opposed to poorly differentiated cells. Interestingly, RNA interference-mediated knockdown of RAC1B decreased expression of the epithelial marker protein E-cadherin, encoded by *CDH1*, and enhanced its TGF-β1-induced downregulation, whereas ectopic overexpression of RAC1B upregulated *CDH1* expression and largely prevented its TGF-β1-induced silencing of *CDH1*. Conversely, knockdown of RAC1B, or deletion of the RAC1B-specific exon 3b by CRISPR/Cas-mediated genomic editing, enhanced basal and TGF-β1-induced upregulation of mesenchymal markers like Vimentin, and EMT-associated transcription factors such as SNAIL and SLUG. Moreover, we demonstrate that knockout of RAC1B enhanced the cells’ migratory activity and derepressed TGF-β1-induced activation of the mitogen-activated protein kinase ERK2. Pharmacological inhibition of ERK1/2 activation in RAC1B-depleted cells rescued cells from the RAC1B knockdown-induced enhancement of cell migration, TGF-β1-induced downregulation of *CDH1*, and upregulation of *SNAI1*. We conclude that RAC1B promotes epithelial gene expression and suppresses mesenchymal gene expression by interfering with TGF-β1-induced MEK-ERK signaling, thereby protecting cells from undergoing EMT and EMT-associated responses like acquisition of cell motility.

## 1. Introduction

The human *RAC1* gene, by alternative splicing, gives rise to two proteins designated Ras-related C3 botulinum toxin substrate 1 (RAC1) and RAC1B. Both proteins belong to the Rho family of monomeric GTPases with RAC1B differing from RAC1 by the presence of an additional exon (exon 3b) of 19 amino acids in length. Biochemically, the in-frame insertion of exon 3b results in an accelerated GDP/GTP-exchange and an impaired GTP-hydrolysis compared to RAC1. In addition, RAC1B differs from RAC1 by the type of upstream activators, binding partners, and downstream effectors/targets, although only few RAC1B-specific target genes have been identified so far. RAC1B has been implicated in tumor progression by its ability to promote cell cycle progression and apoptosis resistance in some cell types, however, its role in other processes driving malignant transformation such as epithelial-mesenchymal transition (EMT), migration/invasion, and metastasis is less clear (for review see [1]).

Pancreatic ductal adenocarcinoma (PDAC) is one of the most malignant tumors with an extremely poor prognosis [2,3]. This is due to its highly metastatic nature and therapy resistance [4] with most patients dying from the consequences of metastatic spread to other organs, particularly to the liver. In order to detach from the primary tumor, become motile, invade surrounding tissues, and eventually colonize distant sites in the host the tumor cells must undergo a process termed EMT. EMT is an evolutionary conserved genetic program that has been adopted by many carcinomas to facilitate invasion and metastasis, cancer stem cell formation, as well as therapy resistance and cancer relapse [5]. In fact, EMT strongly correlates with the systemic aggressiveness of pancreatic tumors [6] and is associated with tumor budding as inferred from association with the EMT marker Vimentin [7].

The tumor-promoting effect of RAC1 is based primarily on its pro-EMT, proinvasive and prometastatic function in several tissues [8,9,10]. In a mouse model of oncogenic Kras(G12D)-induced PDAC, Rac1 was required for early metaplastic changes and neoplasia-associated actin rearrangements in development of pancreatic cancer [11]. Moreover, RAC1 which is hyperactivated in PDAC [12] may contribute to the desmoplastic reaction (a hallmark of PDAC) and the detrimental characteristics of transforming growth factor (TGF)-β1 in advanced-stage disease [13] due to its ability to promote fibrotic signaling by TGF-β1 [14]. While the role of RAC1 as a mediator of EMT is well established, this is not the case for RAC1B. Whereas Rac1b has been reported to promote EMT induced by matrix metalloproteinase 3 (MMP3) in an immortalized mouse mammary epithelial cell line [15,16], our group observed in human PDAC-derived ductal epithelial cells that RAC1B potently inhibited mesenchymal differentiation induced by TGF-β1 [17]. The potential role of RAC1B as an endogenous inhibitor of (TGF-β-dependent) EMT is supported by its potent suppressive effect on basal and TGF-β1-induced cell migration (a hallmark feature of EMT) in various benign and malignant human cell lines of pancreatic and breast origin [17,18,19,20,21].

Earlier studies focused on genes that disturb the epithelial phenotype and promote activation of EMT and mesenchymal differentiation such as RUNX2 [22]. More recent studies have identified a set of yet other genes that establish and maintain an epithelial phenotype in cells and thereby prevent mesenchymal transdifferentiation/EMT such as RUNX1 [23]. The proteins encoded by these genes act as important barriers against tumor growth and malignant transformation. This is exemplified by the cell adhesion molecule E-cadherin which is critical for the maintenance of epithelial tissue structure and is a known tumor suppressor [24,25,26]. In this study, we analyzed how RAC1B impacts epithelial and mesenchymal gene expression in a panel of permanent PDAC-derived cell lines with different differentiation states/phenotypes. We show here that RAC1B (i) is preferentially expressed in benign pancreatic duct epithelial cells, and in well differentiated PDAC cells, (ii) promotes the expression of epithelial genes and protects them from the action of TGF-β which enforces loss of the epithelial phenotype, and (iii) inhibits basal and TGF-β-induced expression of mesenchymal genes, and random cell migration. Moreover, we provide evidence that RAC1B’s effects on TGF-β regulation of E-cadherin and SNAIL, a master regulator of EMT, as well as on cell motility are mediated by suppression of MEK-ERK signaling.

## 2. Material and Methods

### 2.1. Reagents

The following primary antibodies were used: anti-E-cadherin (#610181) and anti-Rac1 (#610650), BD Transduction Laboratories (Heidelberg, Germany), anti-Claudin-7 (#STJ23163), St John’s Laboratory, anti-phospho-ERK1/2 (#4370), anti-GAPDH (14C10), #2118, and anti-Snail (#3895), Cell Signaling Technology (Frankfurt am Main, Germany), anti-HSP90 (F-8), #sc-13119, Santa Cruz Biotechnology (Heidelberg, Germany), anti-RAC1B, #09-271, Merck Millipore (Darmstadt, Germany), anti-Slug (#ab51772), Abcam (Cambridge, UK). HRP-linked anti-rabbit, #7074, and anti-mouse, #7076, secondary antibodies were from Cell Signaling Technology. Recombinant human TGF-β1, #300-023, was provided by ReliaTech (Wolfenbüttel, Germany), and U0126 from Merck/Calbiochem.

### 2.2. Cells

Panc1 human PDAC cells were obtained from the ATCC (Manassas, VA, USA) while the other PDAC cell lines used in this study (Capan1, Capan2, BxPC3, Colo357, MiaPaCa2) were supplied by H. Kalthoff (Kiel, Germany). All cell lines were maintained in RPMI 1640 supplemented with fetal bovine serum (FBS), 1% Penicillin-Streptomycin-Glutamine (Life Technologies, Darmstadt, Germany), and 1% sodium pyruvate (Merck Millipore). The generation and characterization of Panc1 cells stably expressing HA-RAC1B [18], or engineered by CRISPR/Cas9 technology to lack exon 3b of *RAC1* [20], has been described in detail earlier.

### 2.3. Transfection of siRNA and Plasmid DNA

On day 1, cells were seeded into Nunclon^TM^ Delta Surface plates (Nunc, Roskilde, Denmark), and transfected twice, on days 2 and 3, serum-free with either 50 nM of prevalidated siRNAs specific for RAC1B [18] or a scrambled control for 4 h using Lipofectamine 2000 (Panc1 cells) or Lipofectamine RNAiMAX (Capan1 and Colo357 cells) (both from Life Technologies). The procedure for plasmid vectors encoding HA-RAC1B (in β) or MYC-RAC1-L61 (in pRK5) was identical except that cells were transfected only once Afterwards, cells received normal growth medium and were incubated for another 24 or 48 h prior to qPCR, immunoblot analysis, or real-time migration assay.

### 2.4. QRT-PCR Analysis

Total RNA was extracted from Panc1 cells using PeqGold RNAPure from Peqlab (Erlangen, Germany) and purified according to manufacturer’s instructions. For each sample, 2.5 μg RNA were subjected to reverse transcription for 1 h at 37 °C, using 200 U M-MLV Reverse Transcriptase and 2.5 μM random hexamers (Life Technologies) in a total volume of 20 μL. Relative mRNA expression of target genes was quantified by qRT-PCR on an I-Cycler (BioRad, Munich, Germany) using Maxima SYBR Green Mastermix (Thermo Fisher Scientific, Waltham, MA). Data were normalized to the expression of the housekeeping genes GAPDH and/or TATA box-binding protein (TBP). For PCR primer sequences see Appendix A.

### 2.5. Immunoblotting

Cell lysis and immoblotting was essentially performed as described previously [18]. In brief, cells were washed once with ice-cold PBS and lysed with 1× PhosphoSafe lysis buffer (Merck Millipore). Following clearance of the lysates by centrifugation, their total protein concentrations were determined with the DC Protein Assay (BioRad). Equal amounts of proteins were fractionated by polyacrylamide gel electrophoresis on mini-PROTEAN TGX any-kD precast gels (BioRad) and blotted to PVDF membranes. Membranes were blocked with nonfat dry milk or bovine serum albumin and incubated with primary antibodies either for 2 h at RT or overnight at 4 °C. After washing and incubation with HRP-linked secondary antibodies, chemoluminescent detection of proteins was done on a ChemiDoc XRS imaging system (BioRad) with Amersham ECL Prime Detection Reagent (GE Healthcare, Munich, Germany). The signals for the proteins of interest were normalized to those for GAPDH or HSP90 (identity verified by size determination using the SM1841 molecular weight marker from Fermentas/Thermo Fisher Scientific (Appendix A)).

### 2.6. ELISA for TGF-β1

A quantitative determination of bioactive TGF-β1 in culture supernatants was done by enzyme-linked immunosorbent assay (Human/Mouse TGF beta1 ELISA Ready-SET-Go eBioscience/Affymetrix Inc., San Diego, CA, USA). Briefly, culture supernatants of Colo357 and Panc1 cells were appropriately diluted and samples (in duplicate) subjected to enzyme-linked immunosorbent assay (ELISA) according to the manufacturer’s instructions. The detection limit was 25 pg/mL. Data were normalized to the cell number from the respective well.

### 2.7. Migration Assays

The xCELLigence^®^ DP system (ACEA Biosciences, San Diego, CA, USA distributed by OLS, Bremen, Germany) was used to measure in real-time random cell migration of Colo357 and Panc1 cells 48 h after the second round of transfection with RAC1B siRNA (see above). CIM plates-16 (OLS, Bremen, Germany) were prepared for the assay as described in detail previously [18,20] and 60,000 cells were loaded into each well of the upper chamber. Relative migratory activity was recorded in 30 or 60 min intervals for up to 12 h using the RTCA software (version 1.2, ACEA).

### 2.8. Statistical Analysis

Statistical significance was calculated using the unpaired two-tailed Student’s *t* test. Results were considered significant at *p* < 0.05 (*). Higher levels of significance were *p* < 0.01 (**) and *p* < 0.001 (***).

## 3. Results

### 3.1. RAC1B Expression in PDAC-Derived Cell Lines is Associated with a Well Differentiated Epithelial Phenotype

In the course of a previous study we observed that RAC1B protein was more abundant in the well differentiated PDAC-derived cell line, Colo357, than in the poorly differentiated PDAC cell line, Panc1 [18]. This finding led us to assume that RAC1B might be preferentially associated with the epithelial phenotype. To confirm this assumption, we analyzed RAC1B expression in a larger panel of well characterized permanent PDAC cell lines. Among these lines, Capan1, Capan2, and Colo357 cells have been classified as well differentiated (G1), BxPC3 as moderately differentiated (G2), and Panc1 and MiaPaCa2 as poorly differentiated (G3) based on ultrastructural features and population doubling times which were related to the grade of differentiation [27]. Intriguingly, RAC1B was expressed at higher levels in G1 and G2 cell lines compared to G3 cell lines (Figure 1A). Its expression pattern essentially mirrored that of E-cadherin (encoded by *CDH1*) and the tight junction protein Claudin-7 (encoded by *CLDN7*) which both were abundantly expressed in Capan2, BxPC3, and Colo357 cells but only weakly in Panc1 and MiaPaCa2 cells (Figure 1A). High RAC1B and E-cadherin protein expression was also observed in another well differentiated PDAC-derived cell line, Capan1, and in the benign human pancreatic ductal epithelial cell line, H6c7 [28] (Appendix A).

To validate the epithelial phenotype of Capan2, Colo357, and BxPC3 cells and to reveal if E-cadherin and Claudin-7 protein expression corresponds to similar changes in mRNA expression, we processed all five PDAC cell lines for qPCR-based expression analysis of *RAC1B*, *RAC1* as control, *CDH1*, *CLDN7*, and two additional epithelial markers, Epithelial cell adhesion molecule (EpCAM, encoded by *EPCAM*) and Cytokeratin-19 (encoded by *KRT19*). Interestingly, the mRNA levels of *RAC1B*, *CDH1*, *CLDN7*, *EPCAM*, and *KRT19* were all higher in the well differentiated cell lines compared to the poorly differentiated ones (Figure 1B). In contrast to *RAC1B*, mRNA levels of *RAC1* differed only marginally among the cell lines (Figure 1B).

TGF-β1 is a powerful inhibitor of epithelial gene expression. To analyze if PDAC cells are capable of autocrine TGF-β production and if this ability correlates to cellular differentiation stages we measured the concentration of biologically active TGF-β1 in the culture supernatants of well differentiated Colo357 and poorly differentiated Panc1 cells. Notably, the concentration of TGF-β1 was 8-fold higher in supernatants of Panc1 than in those of Colo357 cells (Appendix A). Taken together, we found that RAC1B but not RAC1 expression in PDAC-derived cells is strongly correlated with a well differentiated, epithelial-like phenotype.

### 3.2. RAC1B Knockdown is Associated with a Decrease in CDH1 Expression, an Enhancement of TGF-β1-Induced CDH1 Downregulation, and an Increase in Cell Migration

Prompted by the association of RAC1B expression with higher epithelial differentiation in PDAC-derived cells, we addressed the question if RAC1B is causally involved in maintaining an epithelial phenotype. To do this, we inhibited RAC1B expression in Capan1, Colo357, and Panc1 cells by siRNA-mediated knockdown, targeting exon 3b of *RAC1* [17,18,20,21] followed by measurement of epithelial gene expression. The RAC1B knockdown resulted in downregulation of *CDH1* mRNA in all three cell lines (Figure 2A), and of *CLDN7* and *EPCAM* mRNA in Panc1 cells (Appendix A, for E-cadherin protein expression in Capan1 cells see Appendix A).

The silencing of genes that determine the epithelial phenotype is a crucial event in TGF-β-induced EMT. We therefore asked if RAC1B also impacts downregulation of *CDH1* by TGF-β1. Interestingly, treatment of Panc1 cells with TGF-β1 further enhanced the RAC1B knockdown-induced decrease in E-cadherin protein expression (Figure 2B).

*CDH1* is known to be a potent invasion suppressor gene [24]. Having shown that RAC1B knockdown resulted in silencing of *CDH1*, we reasoned that the reduction in E-cadherin protein should increase the cells’ motility. To verify this assumption, we monitored spontaneous (random) migration in real-time in both control and RAC1B-depleted Colo357 and Panc1 cells. Control Colo357 cells failed to display any migratory activity during the 12-h observation period whereas Panc1 cells showed significant migratory activity (Figure 2C). However, the knockdown of RAC1B rendered Colo357 motile and further enhanced migration of Panc1 cells (Figure 2C). Together, the data so far suggest that RAC1B promotes the expression of epithelial genes and helps to prevent their downregulation by TGF-β1. Moreover, the loss of epithelial gene expression results in derepression of cell migration in both high and low E-cadherin expressing PDAC cells.

### 3.3. Ectopic RAC1B Overexpression is Associated with an Increase in the Expression of Epithelial Genes and Protection from Downregulation by TGF-β

In order to confirm the regulatory interactions between RAC1B and its epithelial target genes, we employed Panc1 cells with transient (Figure 3A) or stable (Figure 3B–D) ectopic overexpression of a HA-tagged version of RAC1B (termed Panc1-HA-RAC1B, Appendix A) [18]. In these cells, the mRNA levels of *CDH1* were increased over those in empty vector control cells (Figure 3A). Analysis of E-cadherin expression in Panc1-HA-RAC1B cells by Western blot analysis showed that protein levels were elevated in three of four clones compared to vector controls (Figure 3B). Interestingly, ectopic expression of a constitutively active RAC1 mutant (Q61L) in Panc1 cells had the opposite effect (Figure 3B). Likewise, 3/3 Panc1-HA-RAC1B clones exhibited more abundant mRNA expression of *CLDN7* (Figure 3C). We then asked if RAC1B when overexpressed can protect cells from losing epithelial gene expression in response to TGF-β1 signaling. Strikingly, the ectopically expressed RAC1B partially protected Panc1 cells from the dramatic downregulation of *CDH1* seen in TGF-β1-treated vector control cells (Figure 3D). The data show that ectopic RAC1B in contrast to ectopic RAC1 can promote epithelial gene expression and protect *CDH1* from being silenced by TGF-β1. This dual function of RAC1B might enable the cell to maintain an (undifferentiated) epithelial phenotype.

### 3.4. RAC1B Inhibits the Expression of Mesenchymal Genes and Their Upregulation by TGF-β

The observation that RAC1B expression was more abundant in well differentiated pancreatic cancer cell lines prompted us to hypothesize that RAC1B promotes epithelial differentiation not only by the upregulation of epithelial marker genes but also by blocking the route to mesenchymal differentiation. Support for this comes from previous studies showing that RAC1B suppressed *SERPINE1* (encoding plasminogen-activator inhibitor 1), *MMP9* (encoding MMP9) [17], and *TGFBRI* (encoding the TGF-β type I receptor activin receptor-like kinase 5) [20]. Here we analyzed if the intermediate filament protein Vimentin (encoded by *VIM*), a marker for EMT [29] associated with pancreatic tumor budding [7], is also targeted for inhibition by RAC1B. Strikingly, knockdown of RAC1B increased TGF-β1-regulated protein (Figure 4A) and mRNA (Appendix A) expression of Vimentin indicating relief from inhibition by RAC1B.

Prompted by the finding that RAC1B downregulated a wide array of mesenchymal genes, we addressed the question of whether RAC1B might be able (i) to target master regulators of mesenchymal differentiation programs such as the EMT-associated transcription factors SNAIL (encoded by *SNAI1*) and SLUG (encoded by *SNAI2*), and (ii) to block their induction by agents that promote mesenchymal (trans)differentiation such as TGF-β. Intriguingly, RAC1B prevented upregulation by TGF-β1 of *SNAI1* at both the protein (Figure 4B) and RNA (Figure 4C) level. We also noted a greatly enhanced derepression of basal and TGF-β1-induced SLUG protein in Panc1 cells with a CRISPR-Cas9 engineered genomic deletion of exon 3b of *RAC1* (Panc1-RAC1B knockout cells, Figure 4D). When compared at the mRNA level, *SNAI2* induction by TGF-β1 was much stronger in the RAC1B knockout vs. knockdown cells (Figure 4E). Finally, we sought to know if ectopically expressed HA-RAC1B affects TGF-β1-induced upregulation of *SNAI2*. The quantification of *SNAI2* mRNA in Panc1-HA-RAC1B clones and vector control cells by qPCR revealed that HA-RAC1B strongly decreased the TGF-β1 effect (Figure 4F). Together, these data show that RAC1B prevents the constitutive and TGF-β-driven expression of mesenchymal genes from different classes including that of master regulators of EMT.

### 3.5. Inhibition of MEK-ERK Signaling by RAC1B Underlies Its Stimulatory Effect on CDH1 and Its Repressive Effect on SNAI1 and Cell Migration

The MEK-ERK pathway is a central driver of (TGF-β-induced) EMT in cancer [30,31] and required for cell migration and invasion in PDAC-derived cells [32]. To analyze the role of ERK activation in RAC1B-mediated control of epithelial and mesenchymal gene expression, we treated Panc1-RAC1B knockdown cells in the absence or presence of TGF-β1 with the MEK inhibitor U0126 and determined the expression of *CDH1* and *SNAI1*. Interestingly, U0126 partially prevented the RAC1B knockdown-induced decrease in basal *CDH1* expression and TGF-β1-induced downregulation of *CDH1* (Figure 5A) as well as the RAC1B knockdown-induced increase in basal *SNAI1* expression and TGF-β1-induced upregulation of *SNAI1* (Figure 5B) at both the protein (Figure 5A,B) and RNA (Appendix A) level.

The ability of U0126 to partially restore *CDH1* and suppress *SNAI1* expression in RAC1B-depleted Panc1 cells led us to hypothesize that inhibition of ERK activation should also be able to relieve the inhibitory effect of RAC1B on cell migration (see Figure 2C). To this end, treatment of Colo357 cells with U0126 potently relieved the migratory activity in RAC1B knockdown cells (Figure 5C).

These results suggested the possibility that RAC1B mediates its stimulatory effect on epithelial genes and its inhibitory effect on mesenchymal genes as well as on cell migration by suppression of MEK-ERK signaling. To study this more directly, we treated Panc1-RAC1B-KO and control cells with TGF-β1 for various times and subjected them to immunoblot detection of phosphorylated forms of ERK1 and ERK2, as indicators of their activation (Figure 5D). Intriguingly, a time-course analysis indicated a biphasic activation pattern with a particularly strong increase in TGF-β1-dependent phosphorylation of ERK2 in RAC1B-KO cells at 15 min and 1 h after TGF-β1 addition. Moreover, ERK2 activation was prolonged, reaching levels of control cells only after the 4 h time point (Figure 5D). The results show that RAC1B promotion of basal *CDH1* expression and prevention of TGF-β1-induced silencing of *CDH1* as well as RAC1B inhibition of basal *SNAI1* expression and prevention of TGF-β1-mediated activation of *SNAI1* involves suppression of MEK-ERK signaling. As a functional consequence of these events, RAC1B acts as a potent inhibitor of EMT and cell migration.

## 4. Discussion

In a previous study evaluating RAC1B protein expression immunohistochemically in tissue biopsies of human PDAC, we found that RAC1B expression correlated significantly with patient survival [18]. Since less malignant carcinomas usually have retained a higher degree of differentiation, this already suggested a possible link between RAC1B and the epithelial phenotype. Unfortunately, due to a limited number of samples for each of the three subgroups (G1, G2, G3) it was not possible to correlate RAC1B with differentiation grade. Here, we studied in more detail the role of RAC1B in maintaining a differentiated cellular phenotype. Using a panel of PDAC-derived cell lines with a high or low grade of differentiation, we observed that RAC1B was more abundant in well differentiated PDAC lines as opposed to poorly differentiated lines (see Figure 1). A higher abundance of RAC1B expression in more differentiated lines was also observed among a panel of eight lung adenocarcinoma cell lines [33]. This suggested the attractive possibility that the correlation of RAC1B expression with the cells’ differentiation grade is not only an epiphenomenon but is functionally involved in maintaining the epithelial phenotype, i.e., by promoting the expression of epithelial genes. To this end, we show that *CDH1* as well as other genes that determine the epithelial phenotype were downregulated upon RAC1B knockdown and upregulated following forced RAC1B overexpression. Our results on RAC1B regulation of *CDH1* contrast with those of another study in human colon cancer cells in which E-cadherin expression was found to be negatively regulated by RAC1B [34]. Whether these conflicting observations are due to tissue-specific differences is currently unclear.

In Panc1 cells, *CDH1* was dramatically downregulated by TGF-β1 and RAC1B knockdown was able to enhance this effect while ectopic overexpression of RAC1B antagonized the TGF-β1-dependent repression of *CDH1*. Besides its role as a promoter of epithelial gene expression we identified RAC1B as a powerful inhibitor of basal and TGF-β1-induced mesenchymal gene expression. This applies for mesenchymal genes with different functions and from different classes such as intermediate filament proteins (Vimentin), receptors involved in TGF-β signaling (ALK5, PAR2), proteinases involved in TGF-β1 activation and matrix turnover (MMP9, PAI-1) as well as master regulators of EMT (SNAIL, SLUG). The coordinated induction of epithelial genes and the concurrent suppression of mesenchymal genes and spontaneous cell migration suggests that RAC1B is a central player and an endogenous RAC1 antagonist in the EMT program (Figure 6).

The molecular basis for the antagonistic effects of RAC1 and RAC1B on EMT and cell motility may be related to different signaling abilities. For instance, RAC1 but not RAC1B is able to interfere with cadherin-mediated adhesion and to destabilize cell-cell contacts, events that require activation of the RAC1 downstream target PAK1. However, RAC1B cannot activate PAK1, suggesting that this may contribute to the inability of RAC1B to disrupt cell-cell contacts and induce junction disassembly [35]. In PDAC cells we observed antagonistic effects of RAC1 and RAC1B on TGF-β1-dependent Smad signaling. Whereas RAC1 promotes Smad activation [36], RAC1B inhibits it [18] and we were able to show that impaired Smad signaling by RAC1B is due to suppression of ALK5 and PAR2 [20,21,37]. In the present study we provide evidence that antagonism of RAC1B and RAC1 extends to regulation of E-cadherin expression, cell migration, and the MEK-ERK signaling pathway. While RAC1 can sustain EMT through activation of MEK1/2 [10], we revealed here a strong inhibitory effect of RAC1B on ERK activation, particularly on ERK2. Experiments involving pharmacological blocking of MEK revealed that suppression of ERK is involved in RAC1B-mediated inhibition of cell migration, prevention of TGF-β1-induced downregulation of *CDH1* and TGF-β1-mediated upregulation of *SNAI1* (see Figure 5). While MEK-ERK signaling has been implicated before in downregulation of *CDH1* by epidermal growth factor, serine protease inhibitor Kazal type 1, and activin B [38,39,40], our observation that TGF-β employs the same pathway to silence *CDH1* is novel. The partial reversal of migratory activity in RAC1B knockdown cells and the TGF-β1-dependent decrease in E-cadherin expression by U0126 may be secondary to U0126-mediated inhibition of TGF-β1-induced upregulation of SNAIL since SNAIL is a potent inhibitor of *CDH1*. These results are in line with recent data showing that RAC1B suppression of PAR2 (a prerequisite for subsequent downregulation of ALK5) could be rescued by inhibition of MEK [21].

In view of the data presented in this study the proposed role of RAC1B as a driver of EMT and tumor progression in PDAC remains questionable and requires careful reassessment. Rac1b has been reported to promote cell migration and EMT induced by MMP3 [15,16], although this conclusion was based on data from only one epithelial cell line of murine breast origin. In work carried out on a panel of lung adenocarcinoma cell lines we found that ectopic RAC1B expression was associated with increased E-cadherin and decreased Vimentin expression and in contrast to RAC1 was unable to induce EMT and invasive activity in an in vivo chorioallantoic invasion model [33]. Moreover, in a transgenic mouse model of lung adenocarcinoma, Rac1b expression alone was insufficient to drive tumor initiation and was not required for K-ras driven cell proliferation [41]. Likewise, in another mouse model of inflammation-induced colon cancer, Rac1b overexpression failed to drive intestinal neoplasia in the absence of an oncogenic driver like Apc [42]. In a study on human PDAC, we found that RAC1B expression was primarily found in the tumor cells and was positively correlated with patient survival [18]. In contrast, a study by Mehner and colleagues found in human PDAC tissue biopsies a strong correlation of MMP3 and RAC1B expression levels in all tumor stages and a significant association of the subcellular distribution of RAC1B with patient outcome [43]. However, the issue of whether RAC1B is causally involved in tumorigenesis or whether its overexpression is merely an epiphenomenon remains open as this study did not provide functional data from cell culture models on the consequences of selective RAC1B knockdown for tumor cell EMT or invasion. Moreover, given the overexpression/hyperactivation of the related RAC1 in PDAC, the ability of MMP3 to induce RAC1 activation [44], and the association of increased activation of MMP3 and upregulation of RAC1 [45], it may well be that RAC1 rather than RAC1B accounted for the putative protumorigenic effect. Based on the results of this study, we speculate that RAC1B overexpression rather than being a driver of tumor progression represents a cellular defense mechanism against TGF-β or inflammatory cytokine-induced EMT and eventual malignant conversion. In support of this hypothesis are data from a murine colon cancer model showing that Rac1b can alleviate carcinogen/acute inflammation-associated carcinogenesis [42].

Tumor budding has been associated with EMT and occurs frequently in pancreatic cancer [7,46]. It is defined as the presence of detached, isolated single cells or small cell clusters scattered in the stroma at the invasive tumor front, essentially reflecting a type of diffusely infiltrative growth. Raf-1 kinase inhibitor protein (RKIP) has been identified as a protein with lower expression in the tumor buds/front compared to the tumor center [47]. Moreover, loss of RKIP is associated with the presence of nodal and/or distant metastases as well as reduced patient survival and hence may be used as a predictor of high-grade tumor budding, tumor aggressiveness, and unfavorable outcome [48]. Remarkably, the loss of RKIP or RAC1B share several consequences in common: the appearance of morphological hallmarks of EMT and an increase in cell motility [17,47], a decrease in E-cadherin and an increase in Vimentin, SNAIL, and SLUG expression [47], as well as derepression of MEK-ERK and p38 MAPK signaling [47,49]. It therefore appears that selective loss of RKIP in the tumor buds can drive tumor progression and eventually metastatic disease. Given the anti-EMT and anti-migratory function of RAC1B, it is tempting to speculate that a similar scenario also holds true for RAC1B.

## 5. Conclusions

The data presented in this study revealed an unexpected dual functional role for RAC1B in pancreatic epithelial cells, (active) maintenance of a differentiated (epithelial) phenotype, and protection from (TGF-β1 driven) EMT and cell motility (Figure 6). High RAC1B, or a high ratio of RAC1B to RAC1 expression, may thus pose a barrier against malignant transformation and together with its potent antimigratory function in vitro, RAC1B qualifies as a genuine tumor suppressor protein. Therefore, strategies to increase its expression over that of its protumorigenic relative, RAC1, may be desirable from a therapeutic perspective.

## Figures and Tables

**Figure 1 cells-08-01569-f001:**
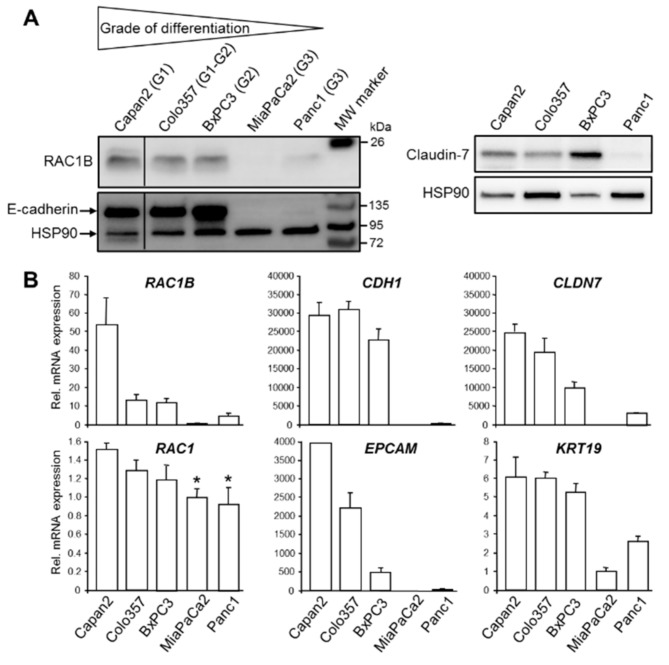
Expression of Ras-related C3 botulinum toxin substrate 1B (RAC1B) in panel of pancreatic ductal adenocarcinoma (PDAC)-derived permanent cell lines correlates with differentiation grade and epithelial phenotype. (**A**) Immunoblot analysis of RAC1B and E-cadherin (left-hand side), and Claudin-7 (right-hand side) in well differentiated (G1, G2) cell lines Capan2, Colo357, and BxPC3, and in the poorly differentiated (G3) cell lines MiaPaCa2 and Panc1. The blots are representative of three experiments with very similar results. The thin lines between lanes 1 and 2 indicate removal of irrelevant lanes. (**B**) QPCR-based detection of RAC1B, RAC1, and various epithelial marker genes, as indicated, in the same cell lines analyzed in (A). Data represent the mean ± SD of three parallel wells and are representative of three experiments. Data are plotted relative to MiaPaCa2 cells set arbitrarily at 1.0. A statistical evaluation of the RAC1 mRNA data revealed that the differences between Capan2 and MiaPaCa2 or Panc1 are significant (*p* < 0.05, unpaired two-tailed Students’ *t* test, indicated by asterisks).

**Figure 2 cells-08-01569-f002:**
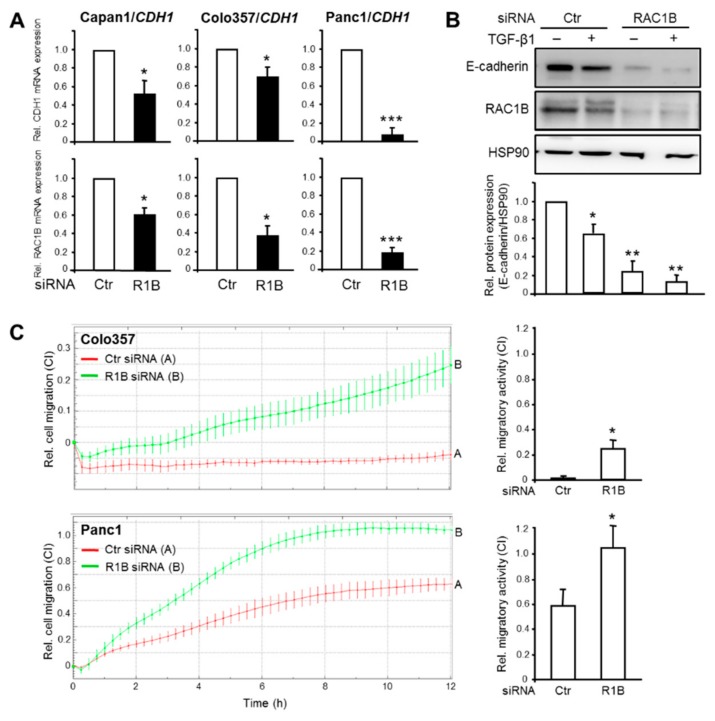
Effect of knockdown of RAC1B on E-cadherin expression and cell migration in PDAC-derived cells. (**A**) Capan1, Colo357, and Panc1 cells were transiently transfected twice with 50 nM each of irrelevant control (Ctr) siRNA or RAC1B (R1B) siRNA and 48 h later processed for RNA isolation and qPCR analysis of the indicated epithelial genes. (**B**) As in (A) except that Panc1 cells were treated with transforming growth factor (TGF)-β1 for 24 h prior to lysis and immunoblot analysis of E-cadherin, RAC1B to verify successful knockdown, and HSP90 as a loading control. The graph below the blot shows results from densitometric quantification of band intensities. Data are the mean ± SD from three independent experiments. The asterisks indicate significant differences relative non-TGF-β-treated, Ctr siRNA transfected cells. (**C**) Kinetics of random cell migration of Colo357 and Panc1 cells after RAC1B knockdown. Colo357 and Panc1 cells were transfected as described in (A) and 48 h later subjected to real-time cell migration assay on a xCELLigence platform (60,000 cells/well). Shown is a representative assay and data are the mean ± SD of three parallel wells. Differences between Ctr siRNA transfected cells (red curve, tracing A) and RAC1B siRNA transfected cells (green curve, tracing B) are first significant at 3:45 (Colo357) and 1:45 (Panc1) and all later time points. The bar graphs to the right depict the differences in migratory activity (expressed by the dimensionless cell index (CI) on the ordinate, please note the different scales) of Ctr vs. RAC1B siRNA transfected cells at the 10-h time point (means ± SD, *n* = 3). Significant differences are indicated by asterisks (Colo357: *p* = 0.002, Panc1: *p* = 0.004). Successful knockdown of RAC1B was verified by immunoblotting (data not shown).

**Figure 3 cells-08-01569-f003:**
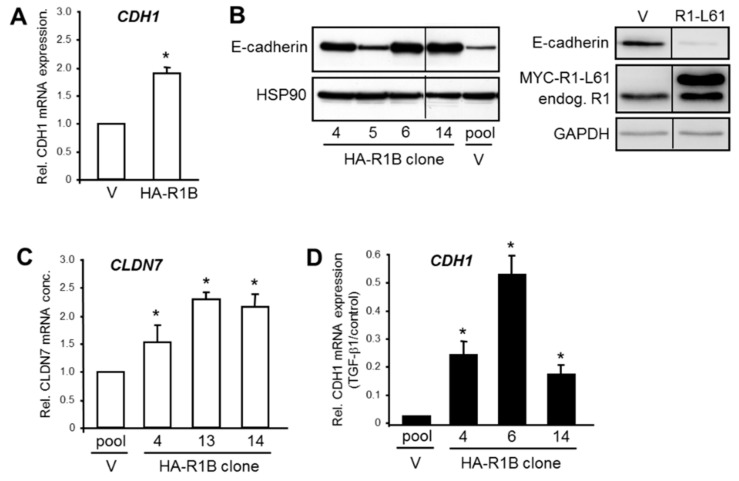
Analysis of basal and TGF-β1-induced E-cadherin expression in Panc1 cells overexpressing HA-RAC1B or constitutively active RAC1. (**A**) QPCR-based analysis of *CDH1*. Panc1 cells transiently transfected with empty pCGN vector (V) or pCGN vector encoding HA-RAC1B (HA-R1B) were analyzed and 48 h after transfection by qPCR for expression of *CDH1*. (**B**) Four individual clones of Panc1-HA-RAC1B cells and vector control cells (left-hand blot), or Panc1 cells transiently transfected with empty pRK5 vector (V) or MYC-RAC1-L61 in pRK5 (R1-L61, right-hand blot), were subjected to immunoblotting for E-cadherin and HSP90 (left-hand blot), or E-cadherin, RAC1, and GAPDH as a loading control (right-hand blot). The blots shown are representative of three experiments with very similar results. The thin lines between lanes indicate removal of irrelevant lanes. Endog—endogenous. (**C**) Three individual clones of Panc1-HA-RAC1B cells and vector controls were subjected to qPCR analysis of *CLDN7*. (**D**) Three individual clones of Panc1-HA-RAC1B cells were treated with TGF-β1 for 48 h and subsequently processed for qPCR analysis of *CDH1*. Data are displayed relative to nontreated control cells set at 1.0. Data in (A,C,D) are the normalized mean ± SD from three independent PCR assays and are plotted relative to vector control cells. Asterisks indicate significance relative to the vector control.

**Figure 4 cells-08-01569-f004:**
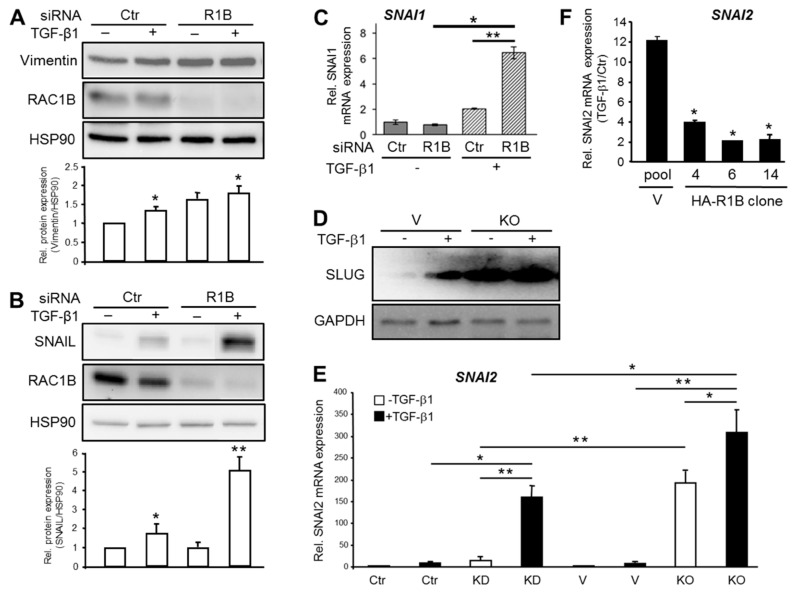
Analysis of basal and TGF-β1-induced expression of mesenchymal genes in Panc1 cells upon knockdown or knockout of RAC1B. (**A**) QPCR-based detection of *VIM* in Panc1-RAC1B-KD cells. Panc1 cells were transiently transfected twice with 50 nM of Ctr siRNA or RAC1B siRNA, stimulated with TGF-β1 for 48 h and subjected to immunoblot analysis of Vimentin, RAC1B, and HSP90. (**B**) As in (A), except that after RAC1B or Ctr siRNA transfection cells were treated with TGF-β1 for 48 h followed by immunoblot analysis of SNAIL and HSP90. The graphs below the blots in (A) and (B) show a quantification of band intensities (normalized mean ± SD from three independent experiments). (**C**) As in (B), except that cells were 24 h after transfection subjected to qPCR analysis of *SNAI1*. Data are the normalized mean ± SD from three experiments. (**D**) Panc1 cells with genomic deletion of exon 3b of *RAC1* (KO) and control cells (V) were treated with TGF-β1 for 24 h prior to immunoblot and analysis of SLUG. The blot shown is representative of three independent experiments. (**E**) As in (D) except that cells were processed for qPCR analysis and compared to RAC1B siRNA-transfected (KD) or Ctr cells. Data are the mean ± SD (*n* = 3). (**F**) Three individual clones of Panc1-HA-RAC1B cells were treated with TGF-β1 for 24 h and subsequently processed for qPCR analysis of *SNAI2*. Data are the mean ± SD from three independent PCR assays and are plotted as TGF-β1-treated over nontreated Ctr cells. Asterisks indicate significance relative to the vector control.

**Figure 5 cells-08-01569-f005:**
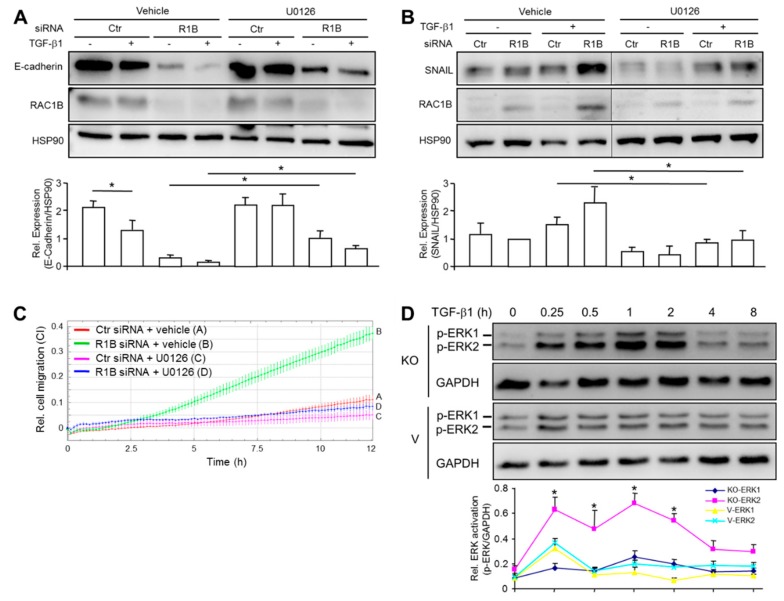
MEK-ERK signaling is involved in TGF-β1 and RAC1B regulation of *CDH1* and *SNAI1*, and Cell Motility. Panc1 cells were transfected twice with 50 nM of an siRNA specific to RAC1B (R1B) or a control (Ctr) siRNA, serum starved overnight and treated with either vehicle (0.1% dimethylsulfoxide), or U0126 (10 μM) in the absence or presence of TGF-β1 (5 ng/mL) for a period of 48 h. Cells were then subjected to immunoblot analysis of E-cadherin (**A**), or SNAIL (**B**), as well as for RAC1B and HSP90. The graphs below the immunoblots show quantification of signal intensities of E-cadherin and SNAIL from three independent experiments (mean ± SD). Data are displayed relative to Ctr siRNA transfected, non-TGF-β1-treated cells. Successful inhibition of ERK1/2 phosphorylation was verified in immunoblots (data not shown). (**C**) Migration assay of Colo357 cells transfected with either RAC1B siRNA or Ctr siRNA and treated with either U0126 (10 µM) or vehicle. Colo357 cells were transfected as described for Panc1 cells in (A) and 48 h later subjected to real-time cell migration assay. The assay shown is representative of three independent assays with very similar results. Data are the mean ± SD of triplicate wells with 60,000 cells/well. Differences between RAC1B siRNA transfected cells + U0126 (blue curve, tracing D) and RAC1B siRNA transfected cells + vehicle (green curve, tracing B) are first significant at 4:15 and all later time points. (**D**) Immunoblot analysis of phospho-ERK1/2 (p-ERK1/2) levels in Panc1-RAC1B-KO cells (KO) and vector controls (V) treated for various times, as indicated, with TGF-β1. The graph underneath the blot depicts quantification of immunoblot data (mean ± SD of triplicate wells). The experiment shown is representative of three assays with very similar results. Asterisks indicate significant differences relative to p-ERK2 abundance in vector control cells for each time point.

**Figure 6 cells-08-01569-f006:**
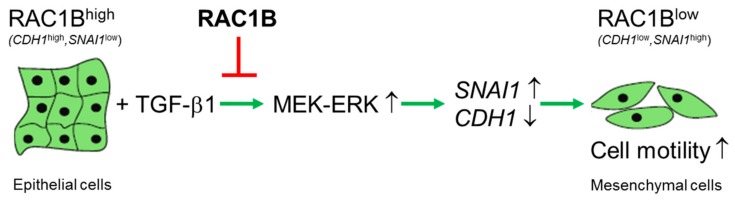
Cartoon illustrating the role of RAC1B in negative regulation of the epithelial-mesenchymal transition (EMT) program in (pancreatic) epithelial cells. Upon exposure to EMT inducers such as TGF-β1, epithelial cells—characterized by high RAC1B and E-cadherin expression, and low SNAIL—activate MEK-ERK signaling leading to downregulation of epithelial marker genes such as *CDH1* and concomitant upregulation of mesenchymal marker genes such as *SNAI1*. The resulting mesenchymal phenotype is characterized by low RAC1B and E-cadherin expression but high SNAIL expression, a spindle-shaped cellular morphology, and a motile phenotype. RAC1B blocks this transition by interfering with TGF-β1-induced activation of the ERK pathway and in particular ERK2. Green arrows indicate activation, while the red line indicates suppression. For details see text.

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
