# Peer review of "RAC1B: A Guardian of the Epithelial Phenotype and Protector Against Epithelial-Mesenchymal Transition"

_cells, 2019, doi:10.3390/cells8121569_

Round 1

Reviewer 1 Report

This study by Zinn et al. reports that Rac1b is more abundant in less differentiated cancer cells, and that reduction of Rac1b promotes EMT, TGFb signaling, and Erk signaling. While all experiments are carried out soundly, the novelty of these results is close to zero as all these findings have already been reported by the same group in earlier publications (Ungefroren et al. 2014, Witte et al. 2017, Ungefroren et al., 2019, Otterbein et al. 2019). Even if the experiments are new, the major conclusions are not.

Suggestions for novel experiments could be fx to check how Rac1b knockout/knockdown is affecting potential Rac1b downstream signaling (Pak1, Pak2?) and Rac1 activation. How will overexpression or ko of Rac1 influence wt and Rac1b ko cells? 

Author Response

Response: We believe that this study contains several novel data that have not previously been reported, i.e. the association of RAC1B expression with the differentiation grade in PDAC-derived cell lines, the positive regulation of E-cadherin and negative regulation of Vimentin by RAC1B, and the involvement of MEK/ERK signaling in these regulatory events. Nevertheless, we thank the reviewer for his suggestions regarding additional experiments the results of which can complement our data. Regarding possible effects on RAC1B downstream signaling it has been shown previously (in COS-7 cells and keratinocytes) that RAC1B is unable to activate full-length PAK (for Ref. see Fiegen et al. J Biol Chem. 2004;279:4743, and doi: 10.1242/jcs.016121). For analysis of the effect of RAC1B knockout/knockdown on RAC1 activation, please see my response to point 3 of Reviewer 3 who suggested the same experiment. As suggested, we ectopically expressed RAC1 (in its constitutively active form, RAC1-Q61L) and found that it decreases E-cadherin expression, (see new immunoblot in Figure 3, panel B, right blot). Moreover, we have performed migration assays with Colo357 and Panc1 cells which show that RAC1B knockdown increases the cells’ migratory activity (see new panel C in Figure 2), an effect that is reversed by treatment with U0126 (see new panel C in Figure 5. The former panel C has become panel D in the revised version).

Reviewer 2 Report

Zinn et al. describe the role of RAC1B in protecting human pancreatic cells from cancer transformation via Epithelial Mesenchymal Transition (EMT), by inhibiting TGFβ-mediated effects in maintaining an epithelial phenotype.

This is a very interesting and well written paper in an important area of research, and introduces a potential basis in better understanding the mechanism of tumor transformation via EMT. However, although it is a well conducted study, there are only two questions as stated below and additional experiments are also needed to resolve these issues.

All the work was done in different, more or less differentiated, pancreatic cancer cells, but, although it is evident the higher expression of RAC1B in differentiated cells than in the lesser ones, that in turn demonstrates the hypothesis of authors, in my opinion some experiments need to be carried out in primary pancreatic epithelial cells or even non-transformed cells to better validate the hypothesis proposed. Authors demonstrated that overexpressed RAC1B acts as inhibitor of TGFβ-mediated EMT. Authors demonstrated this point by adding TGFβ in their cellular models. So, have the authors measured the basal TGFβ secretion in the different pancreatic cells? This referee thinks it should necessary to verify this crucial point and also to analyse if there is a difference in TGFβ production in correlation to differentiation cellular stages.

Author Response

This is a very interesting and well written paper in an important area of research, and introduces a potential basis in better understanding the mechanism of tumor transformation via EMT. However, although it is a well conducted study, there are only two questions as stated below and additional experiments are also needed to resolve these issues.

All the work was done in different, more or less differentiated, pancreatic cancer cells, but, although it is evident the higher expression of RAC1B in differentiated cells than in the lesser ones, that in turn demonstrates the hypothesis of authors, in my opinion some experiments need to be carried out in primary pancreatic epithelial cells or even non-transformed cells to better validate the hypothesis proposed.

Response: We thank the reviewer for this suggestion and have included the benign immortalized human pancreatic ductal epithelial cell line H6c7 in the immunoblot analyses of RAC1B and E-cadherin. As presented in the new Supplementary Figure S1, H6c7 cells show high E-cadherin and high RAC1B expression comparable to that in the G1 cell line Capan2.

Authors demonstrated that overexpressed RAC1B acts as inhibitor of TGFβ-mediated EMT. Authors demonstrated this point by adding TGFβ in their cellular models. So, have the authors measured the basal TGFβ secretion in the different pancreatic cells? This referee thinks it should necessary to verify this crucial point and also to analyse if there is a difference in TGFβ production in correlation to differentiation cellular stages.

Response: We thank the reviewer for bringing up this important issue. We have performed TGF-β1 ELISA in two PDAC lines, Colo357 and Panc1. It turned out that poorly differentiated Panc1 cells secrete 8-fold the amount of well-differentiated Colo357 cells. These data have been included in Supplementary Figure S2. Moreover, we have obtained preliminary evidence that recombinant TGF-b1 downregulates RAC1B mRNA, suggesting the exciting possibility of a causative relationship between high autocrine TGF-b production and low RAC1B expression and vice versa.

Reviewer 3 Report

In this study the authors measure expression levels of Rac1B an isoform splice variant of RhoGTPase, Rac1 in pancreatic cancer cells. They demonstrate using a panel of various pancreatic ductal adenocarcinoma cell lines that expression levels of Rac1B appear to correlate with the differentiation grade of the cancer with loss in poorly differentiated cells. Loss of function experiments using RNAi  to deplete levels and CRISPR KO cells of Rac1B demonstrated loss of Epithelial markers (CDH1) and induction of EMT markers slug and snail in basal activity and in the context of activation by TGF-b1 a potent inducer of EMT.  The authors performed ectopic overexpression experiments and reported Rac1B seems to confer a protective effected to TGF-b1 induced silencing  of (CDH1). They conclude by providing some by biochemical experiments to a potential mechanism to demonstrate that Rac1B supresses signalling via TGF-b1 by interfering with the MEK-ERK signalling pathways.

The is a well thought out paper and shows some new interesting findings about how Rac1B may have a tumor suppressor like function by preserving epithelial like characteristics and how mechanistically be preventing cells undergoing EMT. The data is on the whole of good quality and generally quantified well, however I feel the manuscript would benefit from some additional functional assays to support the biochemical experiments. The manuscript should be considered for publication upon revision of points listed below.

Major points

1

The results are all based on biochemical assays for protein expression/phosphorylation or mRNA levels. It would make a far more convincing manuscript if some of these biochemical experiments were supported by functional assays like a simple cell culture based invasion assay or at least monitoring the migratory behaviour of different cell lines in the loss of function and overexpression experiments. This will help to provide stronger evidence of Rac1B expression being causally linked to tumorgenesis.  

2

It would be further informative to include some immunofluorescent staining of E-cadherin and other Adherens and Tight junction markers in the panel of pancreatic cancer cells to understand the different grades of pancreatict umor cell lines and the effect TGF-b1 stimulation has upon localisation of these proteins.

3

The authors should consider the possible effect Rac1B depletion has on active levels of Rac1 and not just expression levels Rac1, this may support findings that activation of Rac1 maybe an important mechanistic consequence that drives tumor progression upon loss of Rac1B.  I would suggest they monitor the levels of active GTP bound Rac in the depletion/ KO experiments in Figure 2 and 4a.  

Minor points

1

In line 109 the authors state that levels of mRNA of Rac1 differed only marginally among the cell lines (Figure 1B) please could a statistical test be performed to support this claim as the expression levels look different to me.  

In Figure 2A the expression levels of CDH1 are measured in Capan1 cell line yet in Figure 1A we see protein levels of E-cadherin in Capan2 to be consistent can we see the protein levels of the capan1 cell line in Figure 1 A.    

3

 It is not clear what cell line the innunoblot in figure 2B is from can this be added in figure legend  

4

Could the immunoblot of Slug in figure D be replaced with a less saturated one

5

Could it be made clearer in figure 4B (bar graph) what the significant is comparing ? 

6

It would be useful to include the molecular weight markers on all the immunoblots or show the original blots.

Author Response

The is a well thought out paper and shows some new interesting findings about how Rac1B may have a tumor suppressor like function by preserving epithelial like characteristics and how mechanistically be preventing cells undergoing EMT. The data is on the whole of good quality and generally quantified well, however I feel the manuscript would benefit from some additional functional assays to support the biochemical experiments. The manuscript should be considered for publication upon revision of points listed below.

Major points

The results are all based on biochemical assays for protein expression/phosphorylation or mRNA levels. It would make a far more convincing manuscript if some of these biochemical experiments were supported by functional assays like a simple cell culture based invasion assay or at least monitoring the migratory behaviour of different cell lines in the loss of function and overexpression experiments. This will help to provide stronger evidence of Rac1B expression being causally linked to tumorigenesis.

Response: As suggested, we have monitored in real-time the migratory behavior of Colo357 (CDH1high, VIMlow) and Panc1 (CDH1low, VIMhigh) cells after RAC1B knockdown. The results show that i) Panc1 cells have higher basal (random) migratory activity than Colo357 cells and both cell lines respond to RAC1B knockdown with an increase in cell migration consistent with a decrease in E-cadherin and an increase in Vimentin expression. The migration data are presented in Figure 2, new panel C, in the revised version. Moreover, we now show that treatment of RAC1B-depleted Colo357 cells with U0126 reverses the promigratory effect of the RAC1B knockdown (see new panel C in Figure 5. The former panel C has become panel D in the revised version).

It would be further informative to include some immunofluorescent staining of E-cadherin and other Adherens and Tight junction markers in the panel of pancreatic cancer cells to understand the different grades of pancreatic tumor cell lines and the effect TGF-b1 stimulation has upon localisation of these proteins.

Response: This is principally a good idea, however, TGF-β1 can either transcriptionally downregulate CDH1 or change (only) the intracellular localization of E-cadherin protein (away from the cell membrane towards intracellular stores in the absence of transcriptional changes). This is dependent on whether the cells undergo complete or partial EMT (see Ref. Aiello et al. 2018, doi: 10.1016/j.devcel.2018.05.027). Although being highly relevant, the issue of whether RAC1B impacts both or only one of the two forms of EMT is quite complex and we believe that it is beyond the scope of this study. For this reason, we would like these data for a future publication. A comprehensive characterization of the pancreatic cancer cell lines with respect to their differentiation grade can be found in Ref. 27.

The authors should consider the possible effect Rac1B depletion has on active levels of Rac1 and not just expression levels Rac1, this may support findings that activation of Rac1 maybe an important mechanistic consequence that drives tumor progression upon loss of Rac1B.  I would suggest they monitor the levels of active GTP bound Rac in the depletion/ KO experiments in Figure 2 and 4a.

Response: This is certainly a good suggestion. The question of whether RAC1B can suppress the activation of RAC1 is highly relevant as we did observe an increase in RAC1 protein levels following RAC1B knockdown (see Ref. 17). Since the mRNA of RAC1 is also increased there seems to be some kind of crosstalk at the transcriptional or post-transcriptional level, a phenomenon that we are studying currently in a separate project. As requested, we have monitored RAC1 activation in Panc1-RAC1B knockdown cells by immunoprecipitating active GTP bound RAC1 with an anti-active Rac1 antibody followed by immunoblotting with anti-RAC1 antibody. In lysates of RAC1B siRNA transfected cells we were unable to detect GTP-bound RAC1 in contrast to a positive control (Panc1 cells treated with 10 ng/ml EGF for 15 min). Since these results are so far negative in the sense that they do not support the notion of RAC1B driving tumor suppression through inhibition of RAC1 activation, we prefer not to include these data in the revised manuscript version.

Since this study focusses on the regulation of epithelial markers and mesenchymal by RAC1B, we thought it a good idea to confirm the specificity of RAC1B’s effect on E-cadherin by analysing the effects of RAC1 in parallel (a point also suggested by Reviewer 1). Specifically, we have performed transfection experiments in Panc1 cells with a constitutive active RAC1 mutant (Q61L) to show that RAC1, in contrast to RAC1B, inhibits E-cadherin protein expression. These immunoblot data have been included in Figure 3, panel B, right blot. We hope that this Reviewer agrees with us that these data on the role of RAC1 match well with the RAC1B data.

Minor points

In line 109 the authors state that levels of mRNA of Rac1 differed only marginally among the cell lines (Figure 1B) please could a statistical test be performed to support this claim as the expression levels look different to me.

Response: The differences of RAC1 mRNA levels among cell lines are small (ranging from 0.9 to 1.5) compared to those of RAC1B (ranging from 1 to over 50, see different scales at the ordinates). Nevertheless, the difference between Capan2 and MiaPaCa2 on the one hand and Capan2 and Panc1 on the other hand is just significant. This has been indicated in the Figure legend by asterisks.

In Figure 2A the expression levels of CDH1 are measured in Capan1 cell line yet in Figure 1A we see protein levels of E-cadherin in Capan2 to be consistent can we see the protein levels of the capan1 cell line in Figure 1 A.

Response: As requested, protein levels of E-cadherin and RAC1B for Capan1 are shown in the new Supplementary Figure 1. They are in the same range as those for Capan2 and the benign human pancreatic ductal epithelial cell line, H6c7 (see point 1 of Reviewer 2).     

It is not clear what cell line the innunoblot in figure 2B is from can this be added in figure legend.

Response: These data are from Panc1 cells. This has been added in the figure legend.  

Could the immunoblot of Slug in figure D be replaced with a less saturated one.

Response: We certainly agree with the Reviewer that the samples from the KO cells are overexposed. However, the reason for showing this saturated blot was that the signal for the non-TGF-b-treated vector control cells (lane 1) remains visible and allows for an assessment of relative signal intensities among the four samples. Showing a less saturated blot would result in loss of the signal in lane 1. In case this Reviewer agrees with this justification, we would prefer to leave panel 4D as it is.  

Could it be made clearer in figure 4B (bar graph) what the significant is comparing ?

Response: This information was indeed missing. We have now indicated this by horizontal lines.

It would be useful to include the molecular weight markers on all the immunoblots or show the original blots.

Response: As requested, we have included in the Supplementary data file (Figure S7) the original blots from Figures 1A and 4A with signals for the most important proteins analysed in this study (RAC1B, E-cadherin, Vimentin and HSP90), alongside a molecular weight marker.

Reviewer 4 Report

In the article entitled: “RAC1B: a guardian of the epithelial phenotype and protector against epithelial-mesenchymal transition” the authors demonstrated that protein RA1B is a new negative regulator of EMT in pancreatic cancers. The article is very clear and well written. Moreover, the message of this paper is very relevant, but I think that is necessary other experiments to support the role of the RAC1B.

I think that the authors should perform:

-invasion assay (for example zymography);

-migration assay (for example wound healing assay);

-viability assay (MTT assay, SRB assay etc…);

Author Response

In the article entitled: “RAC1B: a guardian of the epithelial phenotype and protector against epithelial-mesenchymal transition” the authors demonstrated that protein RA1B is a new negative regulator of EMT in pancreatic cancers. The article is very clear and well written. Moreover, the message of this paper is very relevant, but I think that is necessary other experiments to support the role of the RAC1B.

I think that the authors should perform: -invasion assay (for example zymography); -migration assay (for example wound healing assay); -viability assay (MTT assay, SRB assay etc…)

Response: As suggested, we have monitored in real-time the migratory activities of two different cell lines, Colo357 (CDH1high, VIMlow) and Panc1 (CDH1low, VIMhigh), after RAC1B knockdown (see also point 1 of Reviewer 3). These data which show that RAC1B knockdown derepresses random cell migration in both cell lines are presented in Figure 2, new panel C, in the revised version. Moreover, we now show that concomitant treatment of RAC1B-depleted Colo357 cells with U0126 can reverse the promigratory effect of the RAC1B knockdown (see new panel C in Figure 5. The former panel C has become panel D in the revised version).

We also performed proliferation assays earlier (see Ref. 18 for RAC1B knockdown cells and Ref. 20 for RAC1B knockout cells). Regarding cell viability/apoptosis, we have never observed the appearance of apoptotic cells following transient RAC1B knockdown or stable RAC1B knockout. However, what we do find are changes in the apoptotic sensitivity of Panc1-RAC1B knockout cells following treatment with the chemotherapeutic drugs, gemcitabine or 5-FU. Evaluating the impact of RAC1B on chemosensitivity is part of a separate project. For this reason, we would prefer not to include these data in the present publication.

Round 2

Reviewer 1 Report

The authors addressed partially the suggestions for improvement. Clearly, understanding the differences between Rac1B and Rac1 effectors is an exciting question to be addressed hopefully by the authors in the future .

Reviewer 2 Report

Zinn et al. have significantly improved the manuscript with the changes made and answered to all this reviewer's comments in an appropriate manner. The manuscript is well written and will be of interest for the scientific community.

Reviewer 3 Report

The Authors have significantly improved the manuscript with new additional functional experiments showing migratory behaviour of cells in the presence and absence of RAC1B as requested,  this  study will now be of  greater interest to the scientific community.  This reviewer thanks them for putting in the effort on this. They have also satisfied all my other major and minor points with their responses this study should now be accepted for publication after a small minor revision (see below )

Just have one minor point about this new data that is not clear in Figure 2C bar graphs on right and side. Could the authors state at what time point in the migration assay the bar graph is plotted I understand a number for when they become significant is stated in legend line 176  ) “are first significant at 3:45 (Colo357) and 1:45 (Panc1)” but this is not the values shown on the bar graph. Could the authors make this clearer so to the readers.

Reviewer 4 Report

The authors have included all requested information. The message of article is better explained.